Subject Areas:
psychology

Keywords:
social discounting, cross-cultural differences, replication, generalizability, sample diversity

Author for correspondence:
Leonid Tiokhin
e-mail: l.tiokhin@tue.nl

†These authors contributed equally to the study.

# Generalizability is not optional: insights from a cross-cultural study of social discounting

Leonid Tiokhin[1,2,†], Joseph Hackman[1,†],
Shirajum Munira[3], Khaleda Jesmin[3]
and Daniel Hruschka[1,†]

[1]School of Human Evolution and Social Change, Arizona State University, Tempe, AZ 85287, USA
[2]Human Technology Interaction Group, Eindhoven University of Technology, IPO 1.24, PO Box 513, 5600 MB, Eindhoven, The Netherlands
[3]LAMB Project for Integrated Health and Development, Parbatipur 5250, Bangladesh

LT, 0000-0001-7333-0383; JH, 0000-0003-4018-7490;
DH, 0000-0003-2188-7782

Current scientific reforms focus more on solutions to the problem of reliability (e.g. direct replications) than generalizability. Here, we use a cross-cultural study of social discounting to illustrate the utility of a complementary focus on generalizability across diverse human populations. Social discounting is the tendency to sacrifice more for socially close individuals—a phenomenon replicated across countries and laboratories. Yet, when adapting a typical protocol to low-literacy, resource-scarce settings in Bangladesh and Indonesia, we find no independent effect of social distance on generosity, despite still documenting this effect among US participants. Several reliability and validity checks suggest that methodological issues alone cannot explain this finding. These results illustrate why we must complement replication efforts with investment in strong checks on generalizability. By failing to do so, we risk developing theories of human nature that reliably explain behaviour among only a thin slice of humanity.

## 1. Introduction

In the course of developing a very sophisticated science of hypothesis testing and experiment, we have almost forgotten the important precursors of these activities ... first, we have to find out what it is that we will be studying, what its properties are, and its generality outside of the laboratory and across cultures.

— Rozin, P. [1, p. 436].

A long-term goal of psychological science is to produce robust and generalizable theories of human nature [1,2]. In recent years, we have become increasingly aware of how inefficiencies in the scientific process obstruct progress towards this goal and may contribute to the unreliability of published research findings [3–9]. This realization has inspired a revolution. Psychologists (along with scholars from diverse disciplines) have united to propose a range of innovative reforms to current scientific practice [3,4]. One central reform has been an increased emphasis on direct replications, studies that recreate the essential elements of previous research to assess the ability of a specific method to generate the same results upon repetition [10]. Direct-replication efforts are rapidly changing the scientific landscape. Large-scale interdisciplinary replication teams are emerging across the globe, scientific journals are increasingly publishing direct replications, and federal governments are beginning to invest resources in replication efforts [11,12].

This cultural shift is long overdue: direct replications are essential for determining the reliability of findings [10,13]. Nonetheless, direct replications are no panacea [14,15]. They often do not reveal boundary conditions for an effect [16,17], the extent to which it replicates under different operationalizations of theoretical constructs [2] or how well it generalizes to different study populations [18]. Recently, Munafò and Smith (scholars who have dedicated their careers towards increasing scientific reliability) expressed concerns that an increased emphasis on direct replication is '... laudable, but insufficient' [15, p. 1] because it underemphasizes the fact that 'strong theories emerge from the synthesis of multiple lines of evidence' [15, p. 3]. In other words, direct replications solve one barrier to developing broadly relevant theories of human nature (i.e. reliability) but do not (and are not designed to) solve others (e.g. generalizability).

## 2. Moving generalizability into the limelight

The vast majority of proposals to improve scientific practice sidestep the issue of generalizability, instead focusing on threats to reproducibility (e.g. *p*-hacking, publication bias, low statistical power, unavailable data and materials, and lack of replication) [3,19]. Those that have directly engaged with concerns about generalizability focus largely on experimental design and statistical analysis. For instance, radical randomization of experimental parameters [20] and crowdsourcing operationalizations of theoretical constructs [21] and analytical choices [22] have all been proposed as ways to reveal how effects vary due to arbitrary choices that researchers make when designing studies. Here, we focus on another longstanding proposal to improve generalizability: increasing sample diversity [18,23].

Social scientists have long worried that convenience samples bias our inferences about human nature by limiting the range of humanity that we study [18,24–28]. Although biases introduced by focusing on members of Western, educated, industrialized, rich and democratic (i.e. 'WEIRD') societies have received the most attention, other convenience samples (e.g. online users, highly educated college students in 'non-WEIRD' societies) present similar problems [29,30]. Despite these concerns, most social-science research continues to rely on participants from these narrow slices of humanity [18,31] and leading proposals for scientific reform scarcely mention the issue of sample diversity ([3], but see [17,23]). It is not clear why worries about unrepresentative participants have not translated into tangible changes to scientific practice, whereas worries about unreliable effects have. Henrich, Heine and Norenzayan published their widely cited 'WEIRD' paper [18] just 1 year before Simmons, Nelson and Simonsohn published 'False-positive psychology' [32], Bem [33] published his infamous pre-cognition paper and Diederik Stapel admitted to fabricating decades worth of psychological data [34]. The latter three events 'drove psychological science into a spiral of methodological introspection' [11, p. 3]. The former seems to have led largely to brief caveats that acknowledge the unrepresentativeness of participants, cite the WEIRD paper, and go about business as usual [26,31,35].

In this paper, we use our recent multi-site investigation of social discounting as one in an emerging set of case studies to illustrate how failing to conduct checks on generalizability across diverse samples can lead to the production of narrow models of human behaviour (for other examples, see [18,36,37]). We do so acknowledging the fact that convenience samples (including WEIRD populations) are often useful: all authors of this paper have relied and continue to rely on convenience samples in our work. We also do not have a special reason for choosing social discounting as a case study, besides the fact that we have conducted social-discounting research and are familiar with the literature. Rather, we suspect that social-discounting research illustrates an issue that will become increasingly relevant to many fields of psychological science: we have good evidence that a phenomenon reliably replicates in a limited set of conditions, but know little about whether it constitutes a general feature of human nature or is a quirk of the narrow range of participants upon which we so heavily rely.

# 3. Social discounting

In psychology, social discounting is defined as the tendency to bear greater costs to benefit socially close individuals than socially distant ones [38]. Specifically, when given the option to sacrifice money (or some other resource) to provide money (or some other resource) to others, people sacrifice substantially more for socially close partners. Over 50 published studies in the last 10 years document this behavioural bias (see https://osf.io/cfkdr/), including a recent pre-registered direct replication (see https://osf.io/fn9am/). The apparent regularity of a hyperbolic relationship between social distance and generosity has led scholars to hypothesize a fundamental relationship with time discounting [38] and investigate its neural basis [39]. Others give social discounting law-like status [40], or describe it as a 'robust phenomenon, with respondents across settings and cultures reliably willing to sacrifice more resources for socially close others relative to distant others' [41, p. 1].

From the perspective of assessing reliability, using the same method to find a recurring hyperbolic relationship between social distance and willingness to sacrifice could be considered a success. Yet, we know little about whether successful replication extends beyond the limited range of participants and methods in these studies. To understand the scope of this potential problem, we reviewed all social-discounting studies that cited Jones and Rachlin's seminal social-discounting paper [38] and used a comparable protocol (see https://osf.io/k8sbg/). Of 43 groups of participants from 21 publications, 40 groups were from the USA and/or university students, with only three groups as exceptions (one from an Indian Mechanical Turk sample [42] and two from one study in Singapore [43]). Even research cited as supporting the cross-cultural reliability of social discounting typically relies on university students [44–46].[1]

# 4. The standard social-discounting protocol

The standard protocol for assessing social discounting was developed with US college undergraduates ([38]; a similar protocol is used in evolutionary psychology to study welfare-trade-off ratios [48]). Typically, it consists of a paper-and-pencil task where participants imagine a list of 100 people closest to themselves. Participants then identify a person (recipient) at a specific location on that list (e.g. #1, #2, #5, #10, #20, #50, #100). For each recipient, participants make several decisions about keeping some amount of money for themselves or transferring some amount to the recipient. In a typical task [41,49,50], participants might choose between option A and option B as follows:

A. $85 for you alone. B. $75 for the #___ person on the list.

A. $75 for you alone. B. $75 for the #___ person on the list.

A. $65 for you alone. B. $75 for the #___ person on the list.

. . .

A. $5 for you alone. B. $75 for the #___ person on the list.

To assess individual generosity, analyses typically calculate the 'crossover point' in the sequence of questions where respondents switch from response A (i.e. the selfish option) to response B (i.e. the generous option). For example, if a participant chooses the selfish option at $85 and $75 but switches to the generous option at $65, her crossover point is $70. This approach expects that participants will switch from generous to selfish only once in a sequence of decisions. Participants that make multiple crossovers are labelled 'inconsistent' and often excluded from analyses [38,41].

To assess ecological and cultural moderators of social discounting, we adapted this social-discounting task [50] to a low-resource, rural, semi-literate setting in Bangladesh, as well as the most commonly studied population in the literature—US college undergraduates. Doing so revealed a drastically different pattern of responding (table 1 and figures 1–3). Unlike US college undergraduates, Bangladeshi participants were not more generous to socially close partners. We were surprised by this finding: based on dozens of prior studies (figure 3), we expected to find at least some degree of social discounting. We then took advantage of an opportunity to run the same task in another non-Western, low-resource setting: rural Indonesia. Again, we found the same pattern. Indonesian participants were

---

[1]One exception is a study that compared social discounting among US college students, urban Chinese and Kenyan herders [47]. Boyer *et al.* found that social distance had a weaker effect on generosity among Kenyan herders compared to the other two groups. However, because this study did not measure social distance directly (instead using culture-specific categories such as 'high-school friend' or 'same-age-set'), it is unclear how to compare its results to those of typical protocols (or even across languages or cultures) and it did not meet our pre-registered exclusion criteria (see https://osf.io/k8sbg/).

**Table 1.** Generosity as a function of social distance, relative need and relatedness. Fixed effects from multilevel model of social distance, recipient need and relatedness regressed on expected sharing. Model controls for correlated observations from the same participant with random effects for each individual and includes random slopes for social distance and recipient need. CI, 95% confidence intervals.

| | USA expected sharing | | Bangladesh expected sharing | | Indonesia expected sharing | |
|---|---|---|---|---|---|---|
| | estimate (CI) | p-value | estimate (CI) | p-value | estimate (CI) | p-value |
| fixed effects | | | | | | |
| (intercept) | 0.71 (0.57, 0.84) | <0.001 | 0.15 (0.07, 0.22) | <0.001 | 0.67 (0.56, 0.78) | <0.001 |
| ln social distance | −0.10 (−0.12, −0.08) | <0.001 | 0.00 (−0.01, 0.01) | 0.677 | −0.01 (−0.03, 0.02) | 0.626 |
| need | | | | | | |
| recipient equally needy | −0.10 (−0.22, 0.02) | 0.121 | −0.07 (−0.15, 0.01) | 0.084 | −0.20 (−0.29, −0.10) | <0.001 |
| recipient less needy | −0.19 (−0.32, −0.07) | 0.004 | −0.13 (−0.21, −0.05) | 0.001 | −0.31 (−0.42, −0.20) | <0.001 |
| relatedness | 0.05 (−0.08, 0.19) | 0.459 | −0.01 (−0.10, 0.08) | 0.857 | 0.12 (−0.00, 0.25) | 0.056 |

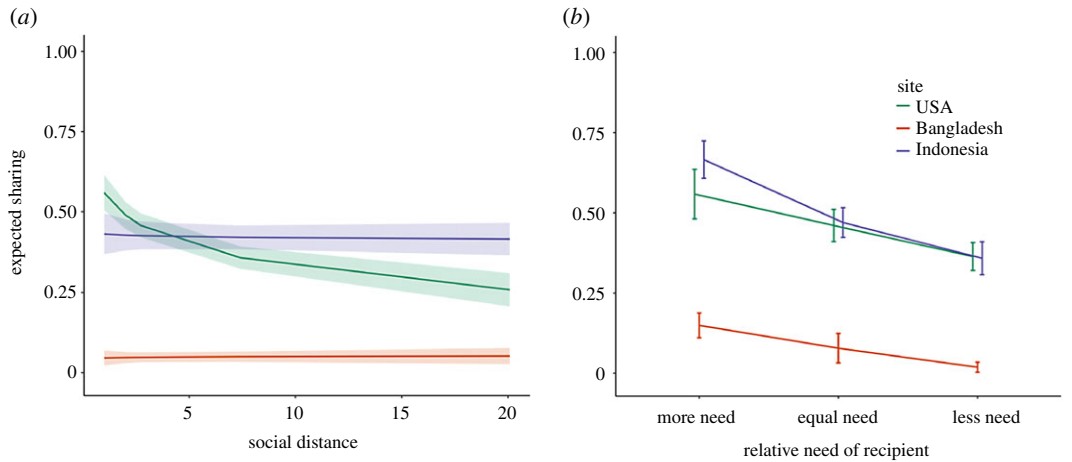

**Figure 1.** Independent effects of social distance and need on generosity. Independent effects of (*a*) social distance and (*b*) relative need on expected sharing. Model estimates from the multilevel model in table 1. Error bars represent 95% confidence intervals.

not more generous to socially close partners. These patterns could not be explained by several common methodological critiques, such as participants failing to understand the protocol, poor measurement of dependent or independent variables, floor effects or low statistical power.

## 5. Methods

Because participants in rural Bangladesh and Indonesia are semi-literate and have varying levels of schooling and experience with typical abstracted paper-and-pencil tasks, adapting the standard protocol to these settings required extensive modifications [51]. These included (i) translation of materials, (ii) piloting and comprehension checks, (iii) identifying locally appropriate idioms for ranking relationships by social distance, (iv) limiting the list of socially close individuals to 20, (v) asking respondents to list and then physically rank cards with 20 socially close individuals rather than asking for abstract rankings of 1, 2, 5, 10 and 20, (vi) modifying how participants identify partners (Bangladesh: choosing among images of all individuals in the village; Indonesia: writing and ranking socially close individuals on notecards), (vii) using an alternative currency (Bangladesh: rice instead of money) to avoid harming ongoing relationships with community members, (viii) presenting choices between selfish and generous options on slips of paper that could be placed in a transparent lottery, (ix) graphic representation of the stakes on slips of paper, and (x) using strategically placed screens to enhance anonymity of decisions. Such challenges are the norm when developing protocols that are meaningful across variable contexts and cultural settings [18,51,52].

Unless otherwise stated, the following adapted protocol was used uniformly across all three sites, including the US sample. Participants made a list of the 20 people that they felt closest to and that did not live in their same household. We identified the idiom most closely aligned with 'social closeness' in Bangladesh and Indonesia from conversations with local respondents about how they describe good relationships where partners care about and help each other (i.e. Bangladesh: *ghonishto*, meaning 'thick/viscous'; Indonesia: *dekat*, meaning 'close'). Participants then sorted the listed individuals in order of how close they felt to each one. The experimenter then selected individuals at five social distances (#1, #2, #5, #10, #20) for the subsequent task. For each of these five individuals, participants made six dichotomous choices (order randomized) between keeping a certain amount of currency for themselves (i.e. selfish option) or giving a certain amount of the currency to that recipient (i.e. generous option). The generous option remained fixed for all choices. The selfish option varied between an amount equal to the generous option to an amount 10% of the generous option. The maximum generous option was scaled to the equivalent of a half-to-full day's wage in each of the contexts (150 Tk in Bangladesh, 25 USD in US undergraduate sample, 50 000 IDR in Indonesia). To assess individual decisions with unfamiliar partners, participants also made decisions between selfish and generous options for an 'unknown person'. Participants in Bangladesh also made decisions between selfish and generous options for an 'acquaintance' in the village.

Each decision was presented as a choice between two paper tickets, one with the selfish option and one with the generous option. For each choice, participants placed their preferred ticket in a small bucket

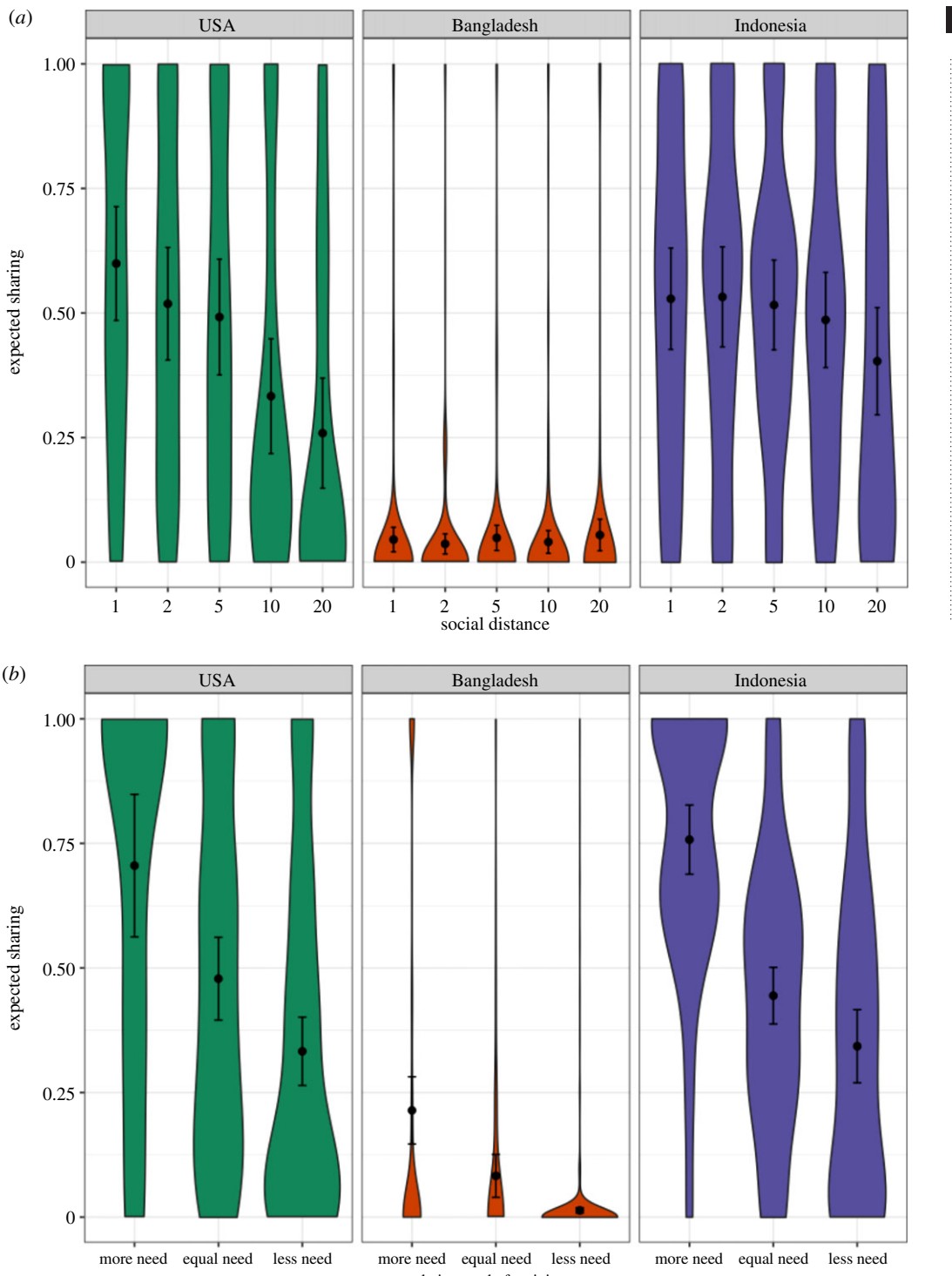

**Figure 2.** Distribution of generosity as a function of social distance and relative need of recipient. Probability density of expected sharing as a function of (*a*) social distance and (*b*) relative need. Dots represent arithmetic means. Error bars represent 95% confidence intervals.

labelled 'lottery', and their non-preferred ticket in a small bucket labelled 'trash'. Participants were instructed that one of the tickets placed in the 'lottery' would be paid out at the end of the experiment, whereas all tickets placed in the 'trash' would be thrown away. All decisions were made behind a screen so that the experimenter was blind to participant decisions. Participants were instructed that their choices were anonymous.

We found high rates of 'inconsistency' in choices (i.e. multiple crossover points for at least one recipient) among participants in both Bangladesh and Indonesia (80% and 100% of participants with

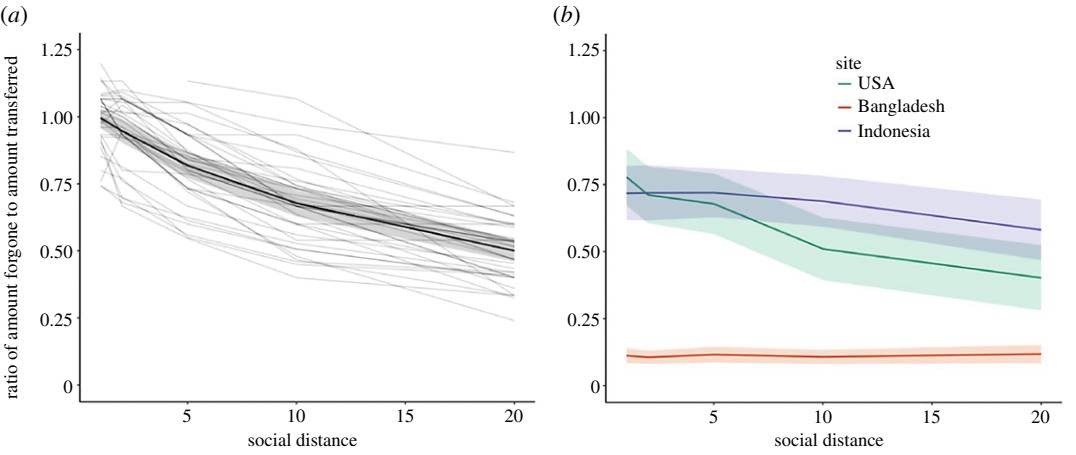

**Figure 3.** Social discounting in prior research and current study. Social discounting, plotted as the ratio of the maximum amount forgone by participants to the amount transferred to recipients. Error bars represent 95% confidence intervals. (*a*) Prior studies (*N* = 39 groups of subjects across 19 publications). (*b*) Current study.

non-zero generosity were inconsistent in Bangladesh and Indonesia, respectively; electronic supplementary material, figure S1). As a consequence, we used a weighted average of respondents' six decisions (henceforth 'expected sharing'). Expected sharing is monotonically increasing with the crossover point when respondents have a single crossover point, does not force exclusion of inconsistent respondents and provides a simple measure of individual differences in generosity (see electronic supplementary material for details and re-analysis using approximated crossover points). Notably, 'inconsistent' respondents did not behave differently from 'consistent' respondents, providing no support for the hypothesis that inconsistent participants failed to comprehend the task (electronic supplementary material, table S17).

Upon finishing the task, participants completed several self-report measures. Participants indicated the relative financial need of the recipient compared to themselves via an ordinal scale (i.e. 'This person is needier than you', 'This person has the same need as you', 'This person is less needy than you'). Participants in the USA and Indonesia also completed the Inclusion of Other in the Self (IOS) Scale [53] as a complementary measure of closeness to the recipient. Bangladeshi participants had difficulty understanding the IOS. As such, in Bangladesh, we developed a protocol using bins of varying distance from the informant, in which villagers could place photos [51,54]. Participants also indicated their age and sex. At the end of the experiment, one choice was randomly selected from the lottery for a payout. If a selfish option was selected, participants received that payment immediately, along with their participation fee. If a generous option was selected, participants only received their participation fee. After all participants completed the task, experimenters paid the appropriate amount to any recipient randomly selected to receive a payout without indicating who it came from.

A total of 284 participants across three sites (USA = 40, Bangladesh = 200, Indonesia = 44) participated in this study (see electronic supplementary material, table S1 for demographic characteristics). In the USA, we recruited 40 participants via e-mails sent to a list of 6000 undergraduates, curated by the Center for Behavior, Institutions, and the Environment. In Bangladesh, we recruited one participant from each of 200 households across four villages in Northwestern Bangladesh [54]. In Indonesia, we recruited 44 participants using opportunity sampling from a single rural settlement (*nagari*) in West Sumatra, near the city of Payakumbuh in the Lima Puluh Kota regency, limiting recruitment to two individuals from the same household.

For the USA and Indonesia, sample size was determined based on the sample size in a previous laboratory study of social discounting [42]. For Bangladesh, sample size was chosen to provide sufficient power (power = 0.80, $\alpha$ = 0.05) to detect a bivariate association between social distance and generosity with a coefficient of determination greater than 0.15.

## 6. Analysis and results

We tested whether generosity declines with increasing social distance by regressing expected sharing on social distance using a multilevel model (table 1). This model controls for genetic relatedness (linear) and

relative need (categorical), and for correlated observations from the same participant with random effects for each individual. It also includes two random slopes: social distance and recipient need (see electronic supplementary material for comparison of alternative models). Because the relationship between money forgone and social distance typically follows a heavy-tailed function, we use the natural log of social distance as a predictor. To adjust for multiple comparisons, we use Bonferroni-adjusted $\alpha$ ($\alpha = 0.004$) levels for all tests of statistical significance based on 12 tests.

For comparability with prior social-discounting research using real stakes, we limit our analysis to decisions for social distances #1, #2, #5, #10 and #20 [49]. In the electronic supplementary material, we also report analyses including generosity towards an 'unknown person' in all three sites (electronic supplementary material, table S15).

Consistent with prior research, we find a strong independent effect of social distance on generosity among US undergraduates, after controlling for need and relatedness ($\beta = -0.10$, 95% CI ($-0.12$, $-0.08$), $p < 0.001$). In contrast, we find no independent effect of social distance on generosity among Bangladeshi ($\beta = 0.00$, 95% CI ($-0.01$, 0.01), $p = 0.677$) or Indonesian ($\beta = -0.01$, 95% CI ($-0.03$, 0.02), $p = 0.626$) participants (table 1 and figure 1a. See electronic supplementary material for sensitivity analyses using different model specifications and BIC-approximated Bayes factors [55]). The independent effects of social distance on generosity in Bangladesh and Indonesia are significantly different from the effect among US undergraduates (($\beta = 0.10$, 95% CI (0.08, 0.12), $p < 0.001$) and ($\beta = 0.10$, 95% CI (0.07, 0.12), $p < 0.001$; electronic supplementary material, table S8) but are not significantly different from each other ($\beta = -0.01$, 95% CI ($-0.03$, 0.02), $p = 0.541$). The maximum plausible effect-size estimates in Bangladesh [$\beta = -0.01$] and Indonesia [$\beta = -0.03$] are several times smaller than the minimum plausible effect-size estimate in the USA [$\beta = -0.08$].

Figure 2 plots the distribution of expected sharing in all sites, as a function of social distance and the relative need of a recipient. This depicts a slight decrease in generosity with increasing social distance among Indonesian participants in the raw data. However, the apparent decrease in generosity with increasing social distance among Indonesian participants is due to a confounding effect of recipients' relative need at varying social distances: we find this effect only when removing relative need as a fixed-effect predictor from the multilevel model in table 1 (electronic supplementary material, tables S5–S8).

This study's finding—that generosity does not increase with decreasing social distance—diverges from well-established findings in social-discounting research. Figure 3 plots our findings against those of comparable social-discounting studies, comprising 39 groups of participants from 19 published articles (for data and pre-registration, see https://osf.io/k8sbg/). For comparability, we calculate the ratio of the maximum amount forgone by participants to the amount transferred to recipients in all studies.

To further probe why social distance did not predict generosity among Bangladeshi and Indonesian participants, we analysed participants' verbal statements about the reasons for their decisions. In all sites, we asked participants to explain their decisions at the end of the task. Bangladeshi and US participants were asked to justify their decisions towards each recipient, whereas Indonesian participants were asked for justifications once, after making all decisions towards all recipients. After reading a subset of statements, we generated a codebook with 13 categories (see electronic supplementary material). Each author independently coded all participant statements in all sites and we resolved conflicting codes via collaborative discussion. We then analysed the subset of codes most relevant to our findings: statements about qualities of relationships and statements about own or recipient need (see https://osf.io/cfkdr/ for complete dataset). Figure 4 plots the proportion of participants who made at least one statement about relationships or need when justifying their decisions. The majority of participants in all sites mentioned need as a justification for their decisions. In contrast, only in the USA did the majority of participants mention relationships. In Bangladesh and Indonesia, 15% and 45% of participants mentioned relationships, respectively. In Indonesia, many respondents mentioned both need and social relationships, even though we did not find an effect of social distance on generosity ($\beta = -0.01$, 95% CI ($-0.03$, 0.02), $p = 0.626$). However, almost all Indonesian participants that mentioned relationships only mentioned the importance of helping family, and Indonesia was the site where we found the largest estimate for the effect of relatedness on generosity ($\beta = 0.12$, 95% CI ($-0.00$, 0.25), $p = 0.056$). Participants' statements appear to correspond closely to their behaviour in the social-discounting task, providing convergent evidence that factors regarding relationship quality have a stronger impact on US participants' behaviour than they do for Bangladeshi and Indonesian participants.

We found that Bangladeshi participants displayed low levels of generosity (160 of 200 participants did not give anything to anyone). To check whether our findings were affected by the inclusion of these participants, we re-ran the same multilevel model on the subset of participants with non-zero

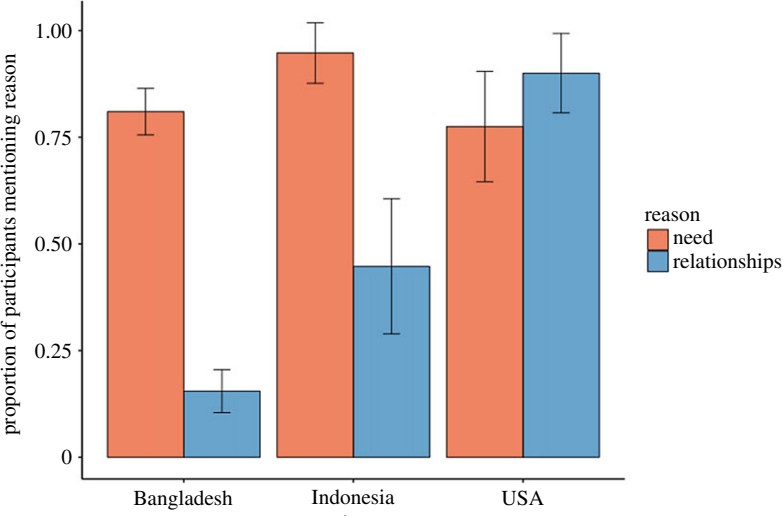

**Figure 4.** Proportion of participants who mentioned need or relationships when explaining their behaviour. Proportion of participants that explained their behaviour in the social-discounting task by making at least one statement about qualities of social relationships (relationships) or own/recipient need (need). Error bars represent 95% confidence intervals.

levels of generosity (USA, $n = 39$; Bangladesh, $n = 35$; Indonesia, $n = 42$). The results were robust to these exclusions (electronic supplementary material, table S14).

One potential reason for the lack of a social-discounting effect in Bangladesh and Indonesia may be that the dependent variable (i.e. amount forgone) was unreliably measured. However, another theoretically relevant covariate—recipient's need relative to participant's need—showed significant independent associations with the dependent variable in all three sites. Specifically, participants were more generous to individuals classified as 'more needy' versus those classified as 'less needy' in all three sites (USA, $\beta = -0.19$, 95% CI $(-0.32, -0.07)$, $p = 0.004$; Bangladesh, $\beta = -0.13$, 95% CI $(-0.21, -0.05)$, $p = 0.001$; Indonesia, $\beta = -0.31$, 95% CI $(-0.42, -0.20)$, $p < 0.001$). This suggests that generosity was measured with sufficient reliability to have substantial and significant associations with at least one theoretically important variable.

Another reason for the lack of a social-discounting effect may be that the independent variable (i.e. social distance) was unreliably measured, despite having shown strong relationships with reported helping in a previous study in the same Bangladesh context [54]. To assess this possibility, we analysed the relationship between social-distance rankings and reports of closeness via the IOS scale (and Bangladesh modification of the IOS), using a multilevel model with random effects for each individual. Participants in the USA ($\beta = -1.23$, 95% CI $(-1.38, -1.08)$, $p < 0.001$), Indonesia ($\beta = -0.86$, 95% CI $(-1.00, -0.72)$, $p < 0.001$) and Bangladesh ($\beta = -0.78$, 95% CI $(-1.00, -0.59)$, $p < 0.001$) reported feeling less close to individuals at greater social distances. Furthermore, participants in the USA ($\beta = -0.03$, 95% CI $(-0.05, -0.01)$, $p < 0.001$), Indonesia ($\beta = -0.06$, 95% CI $(-0.08, -0.05)$, $p < 0.001$) and Bangladesh ($\beta = -0.02$, 95% CI $(-0.02, -0.01)$, $p < 0.001$) were less closely genetically related to individuals at greater social distances.

Another concern in Bangladesh is that of a 'floor' effect on amount forgone. Specifically, if Bangladesh participants had been offered a chance to sacrifice even less than 0.5 kg rice to give their partner 5 kg rice, we may have detected some effect of social distance. However, 0.5 kg rice is already a very low level of sacrifice (10% of what the partner would have received), and there was ample room for examining variation above that low level of potential sacrifice. Moreover, even Indonesian participants who had much higher levels of sacrifice did not exhibit social discounting, independent of other effects (e.g. relative need).

It is possible that social discounting only exists if participants interpret decisions as independent, whereas the large number of inconsistent responses in Bangladesh and Indonesia may indicate that those participants interpreted decisions as cumulative contributions (electronic supplementary material, figure S1). Independent versus cumulative decision-making would represent a previously unknown boundary condition for social discounting. To assess this possibility, we tested for an interaction effect of participant consistency and social distance on expected sharing, using a multilevel model with random effects for each individual. Because all Indonesian participants were inconsistent,

this analysis is limited to the USA and Bangladesh. Among participants with non-zero generosity, there is no evidence that inconsistent participants exhibited less social discounting than consistent participants in the USA ($\beta = -0.01$, 95% CI ($-0.09$, 0.06), $p = 0.687$) or Bangladesh ($\beta = -0.01$, 95% CI ($-0.08$, 0.06), $p = 0.760$) (see electronic supplementary material for BIC-approximated Bayes factors). This suggests that a cumulative interpretation of the task is not sufficient to explain a lack of social discounting.

It remains possible that the current design was insufficiently powered to detect a true relationship between social distance and generosity. To assess this possibility, we used the SIMR [56] package in R [57] to estimate the effect size that our study had 95% power ($\alpha = 0.05$) to detect. We find that the current study had approximately 95% power to detect an effect of [$\beta = -0.018$] in Bangladesh and [$\beta = -0.040$] in Indonesia. This indicates that the current design was sufficiently powered to detect all but the smallest effects. Detecting substantially smaller effects would require samples that are orders of magnitude larger than those of the current study. To illustrate, we calculate the number of participants needed to have 95% power ($\alpha = 0.05$) to detect a true relationship between social distance and generosity, using the estimated effect sizes from table 1. For Bangladesh, detecting an effect of ($\beta = 0.002$) would require approximately 3200 participants. For Indonesia, detecting an effect of ($\beta = -0.006$) would require approximately 4200 participants.

# 7. Discussion

We adapted a common social-discounting protocol, using real stakes, for implementation in three diverse populations: rural Bangladesh, rural Indonesia and US undergraduates. US undergraduates displayed typical patterns of social discounting (i.e. participants incurred substantially greater costs to benefit socially closer individuals). This replicates findings from numerous previous studies. However, we found a fundamental difference between US undergraduates and the two rural populations in the relationship between social distance and generosity: Bangladeshi and Indonesian participants did not exhibit social discounting. Additionally, participants in all sites were more generous to partners categorized as having greater relative need. These findings were consistent with respondents' post-decision rationales for their choices and could not be explained by several potential methodological concerns.

Our protocol differed in several ways from typical protocols (see https://osf.io/cfkdr/ and [51]). These modifications were necessary to implement a study in rural settings with lower literacy rates and participants unfamiliar with typical economic games, and we do not yet know how they affected our results. The fact that our findings among US participants were consistent with past US findings provides evidence that our study retained key aspects of typical protocols. The fact that we documented the same pattern in Bangladesh and Indonesia (i.e. need predicts generosity, but social distance does not) despite different protocols (see Methods) provides convergent evidence that these findings are not artefacts of one specific method [15]. Furthermore, our finding that relative need was the primary determinant of generosity among Indonesian and Bangladeshi participants mirrors findings from other economic games conducted in rural Fiji [58]. Nonetheless, all operationalizations are imperfect, and even seemingly arbitrary differences between protocols can generate dramatically different results [21]. We know embarrassingly little about what construct is captured by the typical measure of generosity used in this study (dichotomous choices between amounts of currency for self versus other) and how well it correlates with alternative operationalizations. This problem is not unique to social discounting (see [59] for a similar issue in studies of risk-preference) and is an important direction for future research.

Despite working with informants to ensure appropriate translations, it is possible that key concepts (e.g. social closeness) were understood differently by participants in Indonesia and Bangladesh than by US undergraduates. However, at least in Bangladesh, extensive interviewing suggests that the local term, *ghonishto*, is the appropriate modifier to describe relationships as intimate, close or familiar. In interviews about their *ghonishto* friends and relationships, people mentioned that one helps them, can rely on them for help and can talk with them about sensitive matters. In Indonesia, translators and local research assistants identified *dekat* as the appropriate term to describe close and intimate relationships. Furthermore, in all three sites, social distance was negatively correlated with another measure of closeness (i.e. IOS) and genetic relatedness (see Analysis and results). This provides evidence that the idioms used in Bangladesh and Indonesia were roughly comparable to US meanings.

Our study absolutely does not support general claims such as 'humans are not more generous to socially close partners' or 'people in Bangladesh are not generous'. Rather, it provides one piece of

evidence against the hypothesis that social discounting is a cross-culturally robust phenomenon. Our study also provides evidence against the hypothesis that generosity is hyperbolically related to social distance [38]: if social distance is unrelated to generosity, then this precludes a hyperbolic relationship with generosity. We can only speculate as to why our results diverge from prior findings. One factor that may affect social discounting is that resource transfers have different meanings across societies [60–62]. For example, informal interviews in Bangladesh revealed that giving without recipient need is a frowned-upon signal of superiority. In Indonesia, individuals are expected to share windfalls and surplus goods (especially with needy others) and are shamed if they don't do so [63]. Norms about whether people should behave according to personal preferences or feelings versus formal social obligations also vary across cultures [64]. It is possible that social discounting only exists when people treat others based largely on individual feelings, as many US undergraduates did in this study (figure 4; see https://osf.io/cfkdr/ for participants' justifications). It is also possible that populations in resource-scarce environments have norms that encourage a focus on relative need over other factors. Future research can assess this possibility by implementing social-discounting protocols among populations that vary in resource scarcity.

Given that norms strongly shape human behaviour across societies, determining their precise effect on social discounting is a promising direction for future research. One approach is to manipulate experimental framing (e.g. instructing participants to make decisions based on 'your duties or obligations towards this person' versus 'your personal feelings towards this person'), as slight changes in framing can dramatically affect behaviour and cognition in cross-cultural settings [65]. However, when norms are internalized, slight framing-changes may be insufficient to change behaviour. In such cases, comparative studies in diverse cultural and ecological settings may be our main window into the scale of human diversity [60].

## 8. Towards a science of reliable and general phenomena

Our cross-cultural investigation of social discounting serves as one case study to illustrate the importance of checks on generalizability across diverse populations as a complement to narrowly focused replication efforts. For any phenomenon, we should strive to conduct those studies that are most valuable to the scientific community. Just as some studies will not be worth replicating [66,67], some will not warrant checks on generalizability (e.g. if an effect has weak empirical support, has a weak theoretical foundation or does not hold up to different operationalizations of theoretical constructs). But checks on generalizability are often essential. As this paper demonstrates, studying phenomena in previously unstudied populations can be useful. However, distinct circumstances warrant distinct approaches to assess generalizability [37]. If we hypothesize that a reliably replicated phenomenon is universal, we should strive to test it among a diverse set of human populations. If we hypothesize that a phenomenon is unique to populations that exhibit some trait (e.g. that only populations with languages that use number words can count exact numbers of large magnitudes), then we can test this hypothesis by using a few target societies as critical tests (e.g. comparing performance on counting tasks in societies with and without number words) [37]. And if we hypothesize that a phenomenon varies as a function of some parameter (e.g. generosity as a function of economic deprivation), but are agnostic as to the source of this variation (e.g. between cultures; between individuals within the same culture), then both within- and between-culture studies can be useful (e.g. economic games among individuals from differentially affluent neighbourhoods within the same city; economics games among individuals from countries with different levels of affluence) [68].

Without seriously considering how to improve the generalizability of our science, we put ourselves at risk. Foremost, we risk generating narrowly replicable effects and theories of human behaviour that tell us little about humanity as a whole [1,2,18]. But we also face another risk: failing to study generalizability in a way that maximizes the information-value of our research. Cross-cultural studies require considerable time and resources (e.g. acquiring permits; starting a field site; establishing trusting relationships with participant populations; learning local languages, norms and ways of life), participants are difficult to recruit (especially for lone field-researchers working in small-scale societies), and fieldwork is physically and psychologically demanding. Two unfortunate consequences are that many behavioural and psychological studies of non-WEIRD populations rely on small numbers of participants and are never directly replicated. Documenting cross-cultural variation may be a necessary first step towards developing generalizable theories of human nature, but it will not be sufficient unless we invest in complementary efforts to ensure that such variation is reliable.

We see several potential ways to address these concerns. One is to increase investment in long-term field sites in non-WEIRD contexts (e.g. [69]) where it is more feasible to acquire large sample sizes and conduct follow-up research. Another is to invest in large-scale collaborative projects (e.g. AHRC Culture and the Mind [70]) that investigate the same phenomenon in diverse contexts [71–73]. One laudable recent initiative, The Psychological Science Accelerator (PSA), has taken the latter approach by developing a distributed network of laboratories across more than 50 countries [23]. The PSA has tremendous potential: cross-cultural research projects will be able to acquire large sample sizes, and laboratory-based experiments could plausibly allow better assessment of how contextual variables influence effect heterogeneity [23]. Nonetheless, the PSA also faces major challenges. For example, if studies rely primarily on college undergraduates in different cultures, they will inevitably recruit participants who are wealthier, more educated, live in settings that are more urban and industrialized, and are otherwise unrepresentative of much of humanity. This could result in a misleading picture of human diversity: consistent findings across different labs may be interpreted as establishing universality, whereas an effect actually depends on a parameter that does not sufficiently vary between laboratories (e.g. exposure to modern society [74]). The extent to which this will be a major issue remains to be determined.

In our continual search for ways to improve our science, we should strongly consider the benefits of increasing investment in tests of generalizability across diverse populations. Although this point is not new [1,2,18,26,75], we believe it deserves far more attention in current discussions on scientific reform. Our plea for stronger checks on generalizability does not imply that replication is not important. On the contrary, an exclusive focus on exploring generalizability without direct replication risks generating a range of interesting effects that are difficult to explain and have questionable reliability [10]. Improving our science requires investment in both determining the reliability of effects and assessing their generalizability across diverse contexts, cultures and populations. Although norms and incentives are shifting in favour of the former, the latter remains woefully undervalued. This needs to change. Only by doing so will we develop models of human nature that are both reliable and broadly relevant.

## 8.1. Constraints on generality

Constraints on generality (COG) statements are a recently proposed addition to empirical papers in Psychology [17]. COGs identify participant samples and assess their representativeness of larger populations, define the specific procedures and materials necessary to conduct a study and identify potential boundary conditions of effects (e.g. historical/temporal factors). Making COGs explicit will increase the transparency of cross-cultural research. COGs also dovetail with recent calls by anthropologists for systematic study of the process by which common protocols are adapted to new cross-cultural settings [51]. Below, we provide a COG statement for this study.

The current study found no evidence for social discounting among participants from rural Bangladesh and Indonesia. We were surprised by this finding, and can only speculate as to the conditions in which it will replicate. We also found that relative need predicted generosity among Bangladeshi, Indonesian and US undergraduate participants. *Participants.* Social distance and generosity: rural, resource-scarce populations in Asia. We have no reason to believe that the effect of social distance or recipient need on participant generosity depends on other characteristics of participants. *Materials.* We have no reason to believe that the results depend on characteristics of the specific materials used in our social-discounting protocol, although the lottery procedure may increase the likelihood of 'inconsistent' responding. *Procedures.* Social closeness should be translated using the same terminology as the current study (i.e. *ghonishto* in Bangladesh; *dekat* in Indonesia). Participants should pass a comprehension check before starting the social-discounting protocol and be ensured that their decisions are anonymous. The effect of relative need on generosity should also occur when decisions are not anonymous (see [58]). *Historical/temporal specificity.* The effect of need on generosity may be driven by cultural norms that promote helping individuals with greater financial need. If so, this effect should not occur when such norms do not exist, or when norms promote exploiting individuals with greater financial need. The effect of social distance on generosity may be affected by cultural norms that sanction giving without need. If so, this effect should not occur when such norms exist. We have no reason to believe that the results depend on other characteristics of the participants, materials or context.

Ethics. Permission to perform this study was granted by the Arizona State University Institutional Review Board: STUDY00004602; STUDY00001752, 1201007249. All participants provided informed consent.

Data accessibility. All data, materials and code are openly available at the Open Science Framework (https://osf.io/cfkdr/). The review of prior social-discounting research was pre-registered (https://osf.io/mw37t/). The direct replication of our laboratory's prior social-discounting study was pre-registered (https://osf.io/fn9am/). The social-discounting study and coding of participant responses were not pre-registered.

Authors' contributions. J.H. and D.H. developed the study concept and design, with input from L.T., S.M. and K.J. L.T., J.H. and D.H. piloted and conducted the study. S.M. and K.J. assisted with collection, curation and interpretation of data in Bangladesh. L.T. analysed the data, with input from J.H. and under the guidance of D.H. L.T. drafted the manuscript, with critical revisions from J.H. and D.H. All authors reviewed and approved the final version of the manuscript.

Competing interests. We have no competing interests.

Funding. D.H. acknowledges support from the National Science Foundation grant no. BCS-1150813, jointly funded by the Programs in Cultural Anthropology, Social Psychology Program and Decision, Risk, and Management Sciences.

Acknowledgements. We are grateful to Alex Danvers, Willem Frankenhuis, Daniel Lakens, Cristine Legare, Sarah Mathew, Robert Boyd and Thomas Morgan for their feedback on previous drafts. We are grateful to two editors and anonymous reviewers at *Psychological Science* for their extensive comments on a previous version of this manuscript, and to editors and reviewers at *Royal Society Open Science* for additional input. We thank all participants. We thank Refi Aksep Sativa and Rika Amelia Black for their assistance in collecting data in Indonesia, and the LAMB project MIS-R team for valuable contributions at all phases of the Bangladesh study.

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
