## [Reviewer comments · Royal Society Open Science]

Review History

RSOS-181386.R0 (Original submission)

Review form: Reviewer 1 (Cristine Legare)

Is the manuscript scientifically sound in its present form?

Yes

Are the interpretations and conclusions justified by the results?

Yes

Is the language acceptable?

Yes

Is it clear how to access all supporting data?

Yes

Do you have any ethical concerns with this paper?

No

Have you any concerns about statistical analyses in this paper?

No

Recommendation?

Accept with minor revision (please list in comments)

Comments to the Author(s)

I rarely review papers this well-written, compelling, and consequential for improving psychological research. This is truly outstanding work, and will make a major contribution to increasing the state of the art in cross-cultural research.

I have a few minor comments and speculations.

1) I was intrigued by the authors' speculation that there is cultural variation in norms surrounding generosity (that giving to those without need is seen as a sign of superiority for example). I think this is worth elaborating on further, for each of the populations. One thing that occurred to me re the results for the U.S. sample, for example, is that it is common for middle class U.S. adults (including college students) to live away from those closest to them. And thus gift giving/generosity may serve a different function (to solidify social bonds in the absence of physical contact).

I think it is also worth commenting (in the final reflections) that research of this kind is tremendously time and labor intensive, and takes longer to publish (utterly worth doing, but this is one of the primary reasons it isn't getting done). The expectations for high volume publication in psychology is actively undermining doing the kind of work this research requires, and the result is that the field is producing a large volume of low quality papers. Why has replication gotten more traction than the "diversity problem"? The field is using the same methods with the same convenience samples to tackle replication.

Lack of training in conducting research outside of WEIRD samples within psychology is a major obstacle as well. Note that the authors are unusual in that they are trained in multiple disciplines (including psychology).

The need for convergent methods is worth mentioning I think. Asking participants to explain is a good example of the benefits of multiple measures, but I think developing multiple methods to study a construct/process is the gold standard (and rarely done in psychological research).

Review form: Reviewer 2 (Chris Chartier)**Is the manuscript scientifically sound in its present form?**

Yes

Are the interpretations and conclusions justified by the results?

Yes

Is the language acceptable?

Yes

Is it clear how to access all supporting data?

Yes

Do you have any ethical concerns with this paper?

No

Have you any concerns about statistical analyses in this paper?

No

Recommendation?

Accept with minor revision (please list in comments)

Comments to the Author(s)

My overall assessment is that this is a strong article worthy of publication. It articulates generalizability concerns clearly and provides a compelling specific example of the non-universality of a well-established finding. I include here a set of questions, critiques, and suggestions for the authors to consider, should a revision of this manuscript be requested.

The authors seem to suggest generalizability should always be a primary concern on par with replicability. I enjoy the “not optional” assertion, but wonder if there is a reasonable ordering of these two research goals, even if just a temporal order. Isn’t it appropriate that replicability be established prior to generalizability? Why attempt to generalize a finding that very well may be spurious even in the original samples tested? Some discussion of this issue may strengthen the paper.

On page 4, starting on line 45. The authors ask why generalizability hasn’t captured the same attention as replicability concerns in the field. I think it’s quite obvious. BEM (discovery of the metaphysically impossible) and Stapel (blatant and widespread fraud) are such glaring and powerful evidence of a sick system. I would argue that generalizability concerns likely can’t have similarly stark demonstrations of fundamental problems (much to the detriment of attention focused on them). This doesn’t make generalizability any less important in my mind, but I think it’s perfectly understandable that replicability has captured more attention given the direct and mind-blowing demonstrations of problems in that sphere. I suggest removing this section.

The authors seem to occasionally conflate (or at least write with a lack of precision about) convenience samples vs. deeper community samples and WEIRD/nonWEIRD cultures. As you argue in the discussion, simply moving into nonWEIRD countries and then collecting convenience samples of college students is only one step in the right direction. I suggest making the distinction more clearly in the introduction and giving each concern to get a bit of independent coverage (highlight problems with convenience samples and WEIRD samples as distinct issues).

I think the authors should discuss further the possible sources of inconsistency in Bangladesh and Indonesia responding? This seems the most problematic element of the empirical portions of the paper, and I was a bit dissatisfied with the lack of attempted explanation. I want to be clear, though: I’m not asking you to HARK! If it’s a real head-scratcher, fine!

The sample size planning is inconsistent the rationale is weak (mirroring earlier studies was given as one justification). This is simply a criticism. Not much can be done now and this isn’t a “fatal” flaw in my mind.

Consider toning down the terminology such as “inconceivable” sample sizes with the PSA. Others have run massive studies. These sample sizes will just become more common now with the PSA.

As I mentioned above, I really love to point about PSA not being a panacea. We need “deep” samples moving away from college students to complement our “broad” samples collected in many countries. No suggestion, just a comment :)

I’m glad you included coverage of COGs. The discussion could highlight how consistent use of COGs could be an excellent mechanism for promoting tests of generalizability. By explicitly specifying COGs, researchers essentially welcome others to follow-up on their work with tests of phenomena in other samples and settings.

Given the focus on making tests of generalizability with non-convenience samples in non-WEIRD countries, I have a nagging concern and general complaint. As the authors themselves point out, collecting these data is very time and resource intensive. One of the best ways to incentivize more of this work is to provide academic incentives for the researchers who help conduct such studies in the local communities. The authors acknowledge the support they received collecting data in Indonesia and mention that the Bangladeshi team provided “valuable contributions at all phases of the Bangladesh study.” I find it odd that none of these individuals appear as authors on this submission. It sounds like some of these individuals contributed across multiple CRediT taxonomy categories, and could be considered for authorship. If we hope to increase “deep” work on generalizability and move beyond convenience samples in nonWEIRD countries, incentivizing the work of our local collaborators should be prioritized.

Summary statement: this article is worth publishing. It is a renewed call for more work on generalizability of psychological science accompanied by an open empirical puzzle, and could serve to spur on further cross-cultural work on this and other phenomena. I suggest acceptance following minor revisions.

Christopher R. Chartier

Decision letter (RSOS-181386.R0)

15-Jan-2019

Dear Dr Tiokhin

On behalf of the Editors, I am pleased to inform you that your Manuscript RSOS-181386 entitled "Generalizability is not optional: Insights from a cross-cultural study of social discounting" has been accepted for publication in Royal Society Open Science subject to minor revision in accordance with the referee suggestions. Please find the referees' comments at the end of this email.

The reviewers and handling editors have recommended publication, but also suggest some minor revisions to your manuscript. Therefore, I invite you to respond to the comments and revise your manuscript.

- Ethics statement

- Data accessibility

If you wish to submit your supporting data or code to Dryad (<http://datadryad.org/>), or modify your current submission to dryad, please use the following link:
<http://datadryad.org/submit?journalID=RSOS&manu=RSOS-181386>

- Competing interests

- Authors' contributions

- Acknowledgements

- Funding statement

Because the schedule for publication is very tight, it is a condition of publication that you submit the revised version of your manuscript before 24-Jan-2019. Please note that the revision deadline

will expire at 00.00am on this date. If you do not think you will be able to meet this date please let me know immediately.

If your manuscript is newly submitted and subsequently accepted for publication, you will be asked to pay the article processing charge, unless you request a waiver and this is approved by

Royal Society Publishing. You can find out more about the charges at <http://rsos.royalsocietypublishing.org/page/charges>. Should you have any queries, please contact openscience@royalsociety.org.

on behalf of Professor Antonia Hamilton (Subject Editor)
openscience@royalsociety.org

Reviewer comments to Author:
Reviewer: 1

Comments to the Author(s)

I rarely review papers this well-written, compelling, and consequential for improving psychological research. This is truly outstanding work, and will make a major contribution to increasing the state of the art in cross-cultural research.

I have a few minor comments and speculations.

1) I was intrigued by the authors' speculation that there is cultural variation in norms surrounding generosity (that giving to those without need is seen as a sign of superiority for example). I think this is worth elaborating on further, for each of the populations. One thing that occurred to me re the results for the U.S. sample, for example, is that it is common for middle class U.S. adults (including college students) to live away from those closest to them. And thus gift giving/generosity may serve a different function (to solidify social bonds in the absence of physical contact).

I think it is also worth commenting (in the final reflections) that research of this kind is tremendously time and labor intensive, and takes longer to publish (utterly worth doing, but this is one of the primary reasons it isn't getting done). The expectations for high volume publication in psychology is actively undermining doing the kind of work this research requires, and the result is that the field is producing a large volume of low quality papers. Why has replication gotten more traction than the "diversity problem"? The field is using the same methods with the same convenience samples to tackle replication.

Lack of training in conducting research outside of WEIRD samples within psychology is a major obstacle as well. Note that the authors are unusual in that they are trained in multiple disciplines (including psychology).

The need for convergent methods is worth mentioning I think. Asking participants to explain is a good example of the benefits of multiple measures, but I think developing multiple methods to study a construct/process is the gold standard (and rarely done in psychological research).

Reviewer: 2

Comments to the Author(s)

My overall assessment is that this is a strong article worthy of publication. It articulates generalizability concerns clearly and provides a compelling specific example of the non-universality of a well-established finding. I include here a set of questions, critiques, and suggestions for the authors to consider, should a revision of this manuscript be requested.

The authors seem to suggest generalizability should always be a primary concern on par with replicability. I enjoy the “not optional” assertion, but wonder if there is a reasonable ordering of these two research goals, even if just a temporal order. Isn’t it appropriate that replicability be established prior to generalizability? Why attempt to generalize a finding that very well may be spurious even in the original samples tested? Some discussion of this issue may strengthen the paper.

On page 4, starting on line 45. The authors ask why generalizability hasn’t captured the same attention as replicability concerns in the field. I think it’s quite obvious. BEM (discovery of the metaphysically impossible) and Stapel (blatant and widespread fraud) are such glaring and powerful evidence of a sick system. I would argue that generalizability concerns likely can’t have similarly stark demonstrations of fundamental problems (much to the detriment of attention focused on them). This doesn’t make generalizability any less important in my mind, but I think it’s perfectly understandable that replicability has captured more attention given the direct and mind-blowing demonstrations of problems in that sphere. I suggest removing this section.

The authors seem to occasionally conflate (or at least write with a lack of precision about) convenience samples vs. deeper community samples and WEIRD/nonWEIRD cultures. As you argue in the discussion, simply moving into nonWEIRD countries and then collecting convenience samples of college students is only one step in the right direction. I suggest making the distinction more clearly in the introduction and giving each concern to get a bit of independent coverage (highlight problems with convenience samples and WEIRD samples as distinct issues).

I think the authors should discuss further the possible sources of inconsistency in Bangladesh and Indonesia responding? This seems the most problematic element of the empirical portions of the paper, and I was a bit dissatisfied with the lack of attempted explanation. I want to be clear, though: I’m not asking you to HARK! If it’s a real head-scratcher, fine!

The sample size planning is inconsistent the rationale is weak (mirroring earlier studies was given as one justification). This is simply a criticism. Not much can be done now and this isn’t a “fatal” flaw in my mind.

Consider toning down the terminology such as “inconceivable” sample sizes with the PSA. Others have run massive studies. These sample sizes will just become more common now with the PSA.

As I mentioned above, I really love to point about PSA not being a panacea. We need “deep” samples moving away from college students to complement our “broad” samples collected in many countries. No suggestion, just a comment :)

I’m glad you included coverage of COGs. The discussion could highlight how consistent use of COGs could be an excellent mechanism for promoting tests of generalizability. By explicitly

specifying COGs, researchers essentially welcome others to follow-up on their work with tests of phenomena in other samples and settings.

Given the focus on making tests of generalizability with non-convenience samples in non-WEIRD countries, I have a nagging concern and general complaint. As the authors themselves point out, collecting these data is very time and resource intensive. One of the best ways to incentivize more of this work is to provide academic incentives for the researchers who help conduct such studies in the local communities. The authors acknowledge the support they received collecting data in Indonesia and mention that the Bangladeshi team provided “valuable contributions at all phases of the Bangladesh study.” I find it odd that none of these individuals appear as authors on this submission. It sounds like some of these individuals contributed across multiple CRediT taxonomy categories, and could be considered for authorship. If we hope to increase “deep” work on generalizability and move beyond convenience samples in nonWEIRD countries, incentivizing the work of our local collaborators should be prioritized.

Summary statement: this article is worth publishing. It is a renewed call for more work on generalizability of psychological science accompanied by an open empirical puzzle, and could serve to spur on further cross-cultural work on this and other phenomena. I suggest acceptance following minor revisions.

Christopher R. Chartier

Author's Response to Decision Letter for (RSOS-181386.R0)

See Appendix A.

Decision letter (RSOS-181386.R1)

05-Feb-2019

Dear Dr Tiokhin,

I am pleased to inform you that your manuscript entitled "Generalizability is not optional: Insights from a cross-cultural study of social discounting" is now accepted for publication in Royal Society Open Science.

on behalf of Professor Antonia Hamilton (Subject Editor)
openscience@royalsociety.org

Appendix A

Dear Dr Tiokhin,

On behalf of the Editors, I am pleased to inform you that your Manuscript RSOS-181386 entitled "Generalizability is not optional: Insights from a cross-cultural study of social discounting" has been accepted for publication in Royal Society Open Science subject to minor revision in accordance with the referee suggestions. Please find the referees' comments at the end of this email.

The reviewers and handling editors have recommended publication, but also suggest some minor revisions to your manuscript. Therefore, I invite you to respond to the comments and revise your manuscript.

Thank you. We have modified our manuscript in accordance with the suggestions of the Reviewers.

Reviewer 1

Comments to the Author(s)

I rarely review papers this well-written, compelling, and consequential for improving psychological research. This is truly outstanding work, and will make a major contribution to increasing the state of the art in cross-cultural research. I have a few minor comments and speculations.

We thank Reviewer 1 for their positive assessment and comments.

1) I was intrigued by the authors' speculation that there is cultural variation in norms surrounding generosity (that giving to those without need is seen as a sign of superiority for example). I think this is worth elaborating on further, for each of the populations. One thing that occurred to me re the results for the U.S. sample, for example, is that it is common for middle class U.S. adults (including college students) to live away from those closest to them. And thus gift giving/generosity may serve a different function (to solidify social bonds in the absence of physical contact).

We have elaborated further on the generosity-norms that may affect social discounting (lines 507-514).

I think it is also worth commenting (in the final reflections) that research of this kind is tremendously time and labor intensive, and takes longer to publish

(utterly worth doing, but this is one of the primary reasons it isn't getting done). The expectations for high volume publication in psychology is actively undermining doing the kind of work this research requires, and the result is that the field is producing a large volume of low quality papers. Why has replication gotten more traction than the "diversity problem"? The field is using the same methods with the same convenience samples to tackle replication.

We agree that this is a problem, and we briefly mention the difficulty of cross-cultural research (550-556). We considered discussing the extent to which changing criteria for judging scholars would incentivize cross-cultural research. However, we feel that this would require substantial additional discussion and is beyond the scope of our paper.

Lack of training in conducting research outside of WEIRD samples within psychology is a major obstacle as well. Note that the authors are unusual in that they are trained in multiple disciplines (including psychology).

We agree completely.

The need for convergent methods is worth mentioning I think. Asking participants to explain is a good example of the benefits of multiple measures, but I think developing multiple methods to study a construct/process is the gold standard (and rarely done in psychological research).

We agree completely and we mention the importance of convergent methods at several points in the paper (lines 68-77, 396-398, 478-480). We have added an additional reference to other experimental economic games conducted in rural Fiji that found a similar finding to our study: sharing is primarily determined by relative need (lines 481-483).

Reviewer: 2

Comments to the Author(s)

My overall assessment is that this is a strong article worthy of publication. It articulates generalizability concerns clearly and provides a compelling specific example of the non-universality of a well-established finding. I include here a set of questions, critiques, and suggestions for the authors to consider, should a revision of this manuscript be requested.

We thank Reviewer 2 for this positive assessment and their comments.

The authors seem to suggest generalizability should always be a primary

concern on par with replicability. I enjoy the “not optional” assertion, but wonder if there is a reasonable ordering of these two research goals, even if just a temporal order. Isn’t it appropriate that replicability be established prior to generalizability? Why attempt to generalize a finding that very well may be spurious even in the original samples tested? Some discussion of this issue may strengthen the paper.

This is an important point. We discuss cases in which generalizability is not worth pursuing and cases where it is more important on lines 529-537. We do not think that there is a general rule for a temporal order generalizability vs establishing reliability, because establishing reliability also depends on the population in which one tests an effect (e.g. an effect may not be reliable in U.S. undergrads but be an important aspect of human nature more broadly).

On page 4, starting on line 45. The authors ask why generalizability hasn’t captured the same attention as replicability concerns in the field. I think it’s quite obvious. BEM (discovery of the metaphysically impossible) and Stapel (blatant and widespread fraud) are such glaring and powerful evidence of a sick system. I would argue that generalizability concerns likely can’t have similarly stark demonstrations of fundamental problems (much to the detriment of attention focused on them). This doesn’t make generalizability any less important in my mind, but I think it’s perfectly understandable that replicability has captured more attention given the direct and mind-blowing demonstrations of problems in that sphere. I suggest removing this section.

Thanks for this suggestion. We in-part share this intuition. However, by one metric (citations) this doesn’t appear to be the case. The Henrich et al. WEIRD paper has been cited over 5000+ times, compared to the 3000+ citations of Simmons et al. False-Positive Psychology. So it seems like the WEIRD paper has caught people’s attention plenty, but that nonetheless little change has happened.

Additionally, Reviewer 1 thought that it was obvious that the reason generalizability hasn’t captured the same attention was not due to a difference in the glaring/powerful evidence, but rather that current incentives for high-volume output explain the difference:

“The expectations for high volume publication in psychology is actively undermining doing the kind of work this research requires, and the result is that the field is producing a large volume of low-quality papers. Why has replication gotten more traction than the “diversity problem”? The field is using the same methods with the same convenience samples to tackle replication.”

Instead of speculating about the causes of the lack of attention to generalizability, we hope that this paragraph encourages productive discussion among psychologists on this topic.

The authors seem to occasionally conflate (or at least write with a lack of precision about) convenience samples vs. deeper community samples and WEIRD/nonWEIRD cultures. As you argue in the discussion, simply moving into nonWEIRD countries and then collecting convenience samples of college students is only one step in the right direction. I suggest making the distinction more clearly in the introduction and giving each concern to get a bit of independent coverage (highlight problems with convenience samples and WEIRD samples as distinct issues).

We have clarified in the introduction that convenience samples are not synonymous with WEIRD samples. As the reviewer alludes to, there is also an important problem of deeper engagement with communities and situating findings in their context. This is an important point, but is not the main point of this paper, and we worry that it will distract from the discussion of generalizability.

I think the authors should discuss further the possible sources of inconsistency in Bangladesh and Indonesia responding? This seems the most problematic element of the empirical portions of the paper, and I was a bit dissatisfied with the lack of attempted explanation. I want to be clear, though: I'm not asking you to HARK! If it's a real head-scratcher, fine!

We discussed this substantially in the SI (see "Inconsistent Responding Across Sites) and have clarified the discussion of this in the main text (lines 437 – 449).

The sample size planning is inconsistent the rationale is weak (mirroring earlier studies was given as one justification). This is simply a criticism. Not much can be done now and this isn't a "fatal" flaw in my mind.

We agree, but we chose to be transparent about the actual reasons for choosing samples of varying sizes instead of generating a post-hoc rationale.

Consider toning down the terminology such as "inconceivable" sample sizes with the PSA. Others have run massive studies. These sample sizes will just become more common now with the PSA.

We have changed "previously inconceivable" to "large".

As I mentioned above, I really love to point about PSA not being a panacea.

We need “deep” samples moving away from college students to complement our “broad” samples collected in many countries. No suggestion, just a comment :)

Thank you for this comment.

I’m glad you included coverage of COGs. The discussion could highlight how consistent use of COGs could be an excellent mechanism for promoting tests of generalizability. By explicitly specifying COGs, researchers essentially welcome others to follow-up on their work with tests of phenomena in other samples and settings.

We have now explained what COGs are and why they are useful in the main text. We have also moved our COG statement into the main text (lines 589 – 616).

Given the focus on making tests of generalizability with non-convenience samples in non-WEIRD countries, I have a nagging concern and general complaint. As the authors themselves point out, collecting these data is very time and resource intensive. One of the best ways to incentivize more of this work is to provide academic incentives for the researchers who help conduct such studies in the local communities. The authors acknowledge the support they received collecting data in Indonesia and mention that the Bangladeshi team provided “valuable contributions at all phases of the Bangladesh study.” I find it odd that none of these individuals appear as authors on this submission. It sounds like some of these individual contributed across multiple CRediT taxonomy categories, and could be considered for authorship. If we hope to increase “deep” work on generalizability and move beyond convenience samples in nonWEIRD countries, incentivizing the work of our local collaborators should be prioritized.

We agree that adding local contributors as authors is one way to incentivize their engagement in such research. Our research group has included the local contributors from Bangladesh on papers that have focused on research at that site:

Hruschka, D. J., Munira, S., Jesmin, K., Hackman, J., & Tiokhin, L. (2018). Learning from failures of protocol in cross-cultural research. *Proceedings of the National Academy of Sciences*, 115(45), 11428-11434.

Hackman, J., Munira, S., Jasmin, K., & Hruschka, D. (2017). Revisiting psychological mechanisms in the anthropology of altruism. *Human Nature*, 28(1), 76-91.

In retrospect, it was a mistake not to include the local contributors in Bangladesh as co-authors on this paper. We thank Reviewer 2 for alerting us to this. The local contributors in Bangladesh have now reviewed the manuscript and given their approval to be added as co-authors. We have also amended the contributions section to reflect each author's contribution.

Summary statement: this article is worth publishing. It is a renewed call for more work on generalizability of psychological science accompanied by an open empirical puzzle, and could serve to spur on further cross-cultural work on this and other phenomena. I suggest acceptance following minor revisions.